# Unsupervised monocular depth estimation with omnidirectional camera for 3D reconstruction of grape berries in the wild

**Yasuto Tamura[1], Yuzuko Utsumi[2]\*, Yuka Miwa[3], Masakazu Iwamura[2], Koichi Kise[2]**

**1** College of Engineering, Osaka Prefecture University, Sakai, Japan, **2** Graduate School of Informatics, Osaka Metropolitan University, Sakai, Japan, **3** Local Incorporated Administrative Agency Research Institute of Environment, Agriculture and Fisheries, Osaka Prefecture, Habikino, Japan

\* yuzuko@omu.ac.jp

## Abstract

Japanese table grapes are quite expensive because their production is highly labor-intensive. In particular, grape berry pruning is a labor-intensive task performed to produce grapes with desirable characteristics. Because it is considered difficult to master, it is desirable to assist new entrants by using information technology to show the recommended berries to cut. In this research, we aim to build a system that identifies which grape berries should be removed during the pruning process. To realize this, the 3D positions of individual grape berries need to be estimated. Our environmental restriction is that bunches hang from trellises at a height of about 1.6 meters in the grape orchards outside. It is hard to use depth sensors in such circumstances, and using an omnidirectional camera with a wide field of view is desired for the convenience of shooting videos. Obtaining 3D information of grape berries from videos is challenging because they have textureless surfaces, highly symmetric shapes, and crowded arrangements. For these reasons, it is hard to use conventional 3D reconstruction methods, which rely on matching local unique features. To satisfy the practical constraints of this task, we extend a deep learning-based unsupervised monocular depth estimation method to an omnidirectional camera and propose using it. Our experiments demonstrate the effectiveness of the proposed method for estimating the 3D positions of grape berries in the wild.

## Introduction

Grapes are cultivated all over the world [1]. Although wine grapes are the most popular in most countries, table grapes account for more than 90% of the grape production in Japan [2]. One characteristic of Japanese table grapes is their high price [2]. Japanese people have a custom of sending fruit as gifts. For this reason, not only grapes but all kinds of fruit are expensive and perfect in both taste and appearance. Although it is an extreme example, there was a time when a bunch of grapes was auctioned for 9,700 US dollars [3]. Major reasons for this include that table grapes in Japan require more care to maintain their quality than wine grapes, and

**Funding:** This study is supported by the following organizations all awarded to Y.U. Osaka Prefectural Credit Federation of Agricultural Co-operatives Industry-academia Collaborative Research Support Project, https://www.jabankosaka.or.jp. The Telecommunication Advancement Foundation Research and Investigation Grant, https://www.taf.or.jp, Tateisi Science and Technology Foundation Research Grant (A), https://www.tateisi-f.org and Nippon Life Insurance Foundation, https://www.nihonseimei-zaidan.or.jp. The funders did not play any role in the study design, data collection and analysis, decision to publish, or preparation of the manuscript.

**Competing interests:** The authors have declared that no competing interests exist.

their cultivation is highly labor-intensive [2]. Among various processes, pruning berries (sometimes called thinning) is a special burden on grape farmers.

Pruning berries is carried out at an early stage of grape growth, after young berries have borne, to produce an optimal spacing between berries to accommodate further growth. This task is important to obtain berries that meet sufficient commercial standards of berry size and sweetness, and also to mold grapes into inverted triangle shapes [4]. Several criteria are used to select the berries to be cut, such as the space between adjacent berries and the total number of berries within each part of a grape bunch. In addition, pruning must be performed quickly because the task must be completed in all the bunches in the grape orchards within a certain period of cultivation. Due to its complexity and sensitivity to time, pruning generally takes a long time to master [5]. In Japan, nurturing new entrants is a critical issue to maintain agricultural workers because the number of agricultural workers has gradually decreased, which has been caused by the effects of the declining birth rate and the aging population, as well as the dislike of hard labor in agriculture [6]. Therefore, we consider building a system that automatically presents the berries that should be pruned out to assist the workers in mastering the task.

To select grape berries for removal, the 3D positions of all berries in a bunch need to be estimated. There are two issues to be addressed when measuring a bunch of grapes. The first issue is sensing bunches in the field in a non-destructive manner because it is necessary to continue growing bunches. The other issue is a constraint on the field conditions. Unlike wine grape cultivation in vineyards, with Japanese table grape cultivation, farmers need to work in confined spaces where grape bunches hang from horizontal trellises at a height of about 1.6 meters, as shown in Fig 1. Such circumstances limit the sensing methods.

One strategy to obtain this 3D positional information is using depth sensors. There are conventional studies that use relatively large sensors based on active stereo methods, in which the scanner emits coded light and detects its reflection from the surface of the target object in indoor environments [7–9]. In those studies, since the grapes are measured after being picked, these measurements are destructive. Moreover, the sensors cannot be used fields because strong sunlight interferes with the light emitted by the scanner. Therefore, we cannot apply depth sensors to measure the bunches.

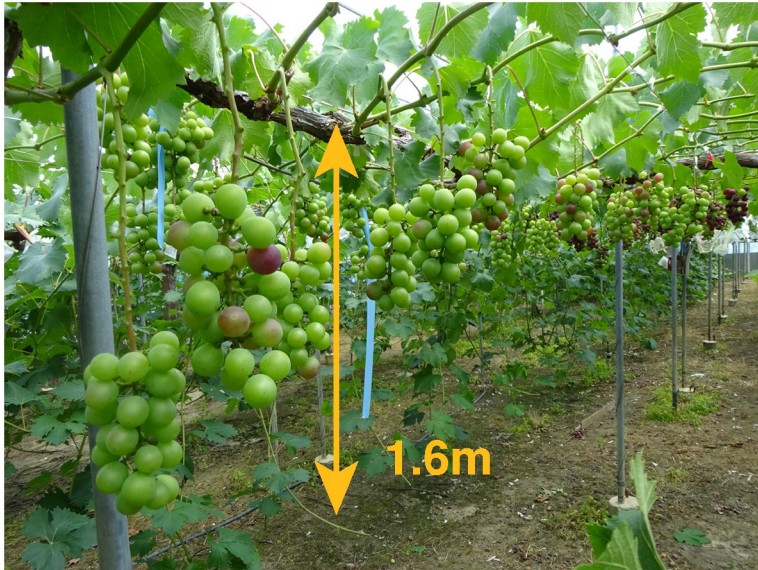

**Fig 1. An example of general Japanese table grape fields targeting this research.**

Another strategy is using passive stereo methods, which do not rely on light emission from the scanner, for a monocular RGB camera, where RGB is short for red, green, and blue. Using a camera, it is possible to capture grape bunches without destruction. In addition, although it is not a bunch of grapes, passive stereo methods have succeeded in reconstructing the 3D shape of a plant taken outdoors [10]. Therefore, in this study, we use an RGB camera to capture the grape bunches and adopt a passive stereo method to estimate the 3D positions to realize non-distractive measurements outdoors. When trying to perform 3D reconstruction using images, it is necessary to capture the bunch of grapes within the frame. As mentioned above, the grape field we are considering is low in height. Thus, it is difficult to capture a video of a whole single grape bunch using a general perspective projection camera with a narrow field of view while avoiding the bunch framing out and bumping into other bunches. For these reasons, in our research, we use one of two lenses of an omnidirectional camera with a wide field of view (i.e., the field of view is 180 degrees).

There is also an issue that needs to be overcome when estimating the 3D positions using images; the grape berries differ from the objects targeted in conventional passive stereo methods. Generally, determining the 3D information of objects captured using a monocular camera requires matching the corresponding points of the objects in different frames of a video so that we can estimate the relative camera pose between the frames. Representative methods that can do this include Visual Simultaneous Localization and Mapping (Visual SLAM) [11–13] and Structure from Motion (SfM) [14]. These methods find corresponding points in different frames by detecting feature points and matching them using the texture around the feature points. However, since grapes are mostly textureless, feature points cannot be detected adequately. Thus, inter-frame correspondence cannot be identified. To tackle such difficulties with textureless objects, alternative methods have been proposed, such as matching of silhouettes [15, 16] or surface curves [17]. [18] conducts 3D reconstruction of textureless and homogeneous objects, such as plant leaves and balloons, using epipolar geometry. However, grape berries are crowded and overlapping, and their shapes are highly symmetric and homogeneous, factors that are likely to lead to failure in berry matching. Therefore, the above 3D reconstruction methods are not suitable for grapes.

A recently developed method, unsupervised monocular depth estimation [19–21], involves the estimation of the relative camera pose between two frames and 3D information (that is, a depth map) using deep learning techniques. In this method, neural network models learn correspondences between consecutive frames. Then, given two adjacent frames as input, the models return the relative camera pose between them and the depth map of each frame. Unlike conventional methods that calculate correspondences between frames based on feature points, silhouettes, or the distribution of pixel luminance values, unsupervised monocular depth estimation calculates the correspondences of entire frames using neural networks. Thanks to the strong expressivity of neural networks, this method is effective for the 3D reconstruction of textureless objects such as roads in scenes of the city taken by the in-vehicle camera [19–21]. Therefore, this method would be effective for grape berries.

However, as far as we know, many unsupervised monocular depth estimation methods assume the use of pinhole cameras in training the models. To show why the camera model is important, we will explain in more detail how to train the network. Given two frame images of a video, if the depth map of an image and the relative camera position between the two images are correctly estimated, we must be able to successfully project the pixel values from one image onto the other without error. Therefore, the conventional methods train the models so that this error becomes small. This training method is called differentiable depth-image-based rendering (differentiable DIBR) [19–21]. When projecting the pixel values, we need to take into account the camera model, which is the reason why the camera model is important. Although

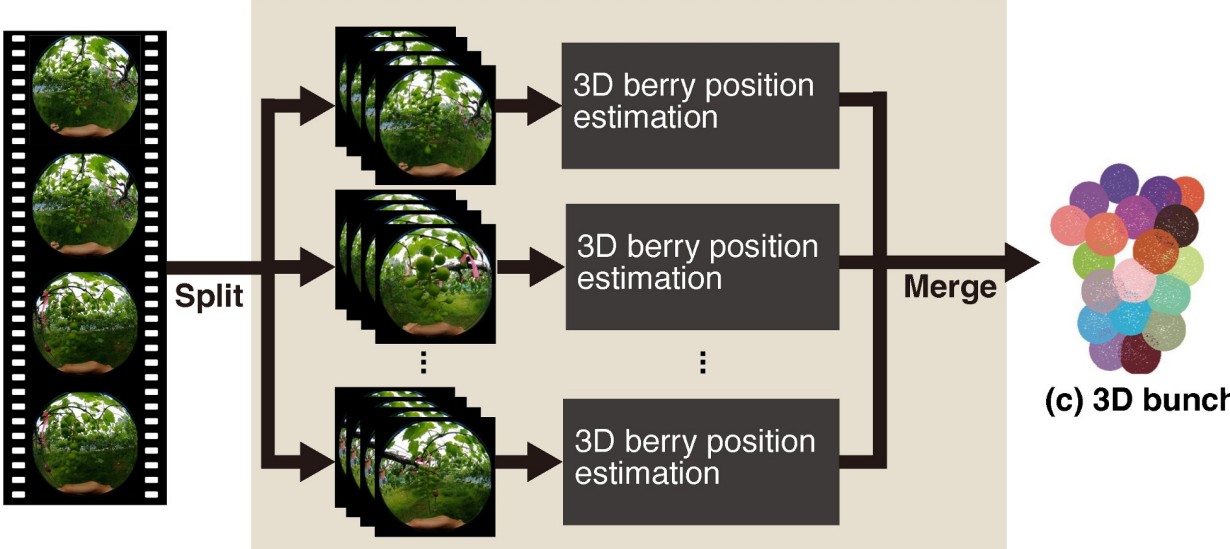

**Fig 2. The procedure of estimating 3D positions of a whole grape bunch.** (a) The input video is taken with an omnidirectional camera. (b) The input video is split into several parts, and 3D reconstruction is performed for each part. The results are integrated to obtain (c) the 3D bunch. The details of the beige rectangle are shown in Fig 6.

there are some conventional methods that address this issue, they have their own drawbacks. Some depth estimation methods have been proposed for wide field-of-view cameras, such as omnidirectional cameras. [22, 23] use deep learning, but they need ground truth depth data for training. [24] requires synthetic stereo data with known baselines of a stereo camera for training (where a baseline is the distance between two cameras), instead of ground truth depth data. Although [25] proposes an unsupervised monocular depth estimation method that can be applied to any camera model, compared to standard monocular depth estimation methods, it requires additional networks to model arbitrary camera models.

Therefore, similar to [26], we extend the differential DIBR of pinhole camera models to omnidirectional cameras and apply it to directly estimating the relative camera poses and the 3D positions of textureless grape berries (see Fig 2). Thanks to the estimation of depth and relative camera pose, grape berries can be tracked across frames if we use a method that estimates the 2D positions of grape berries together. Our experiments demonstrate the effectiveness of the proposed method in estimating the positions of grape berries.

## Materials and methods

### Materials

Grape bunch videos were captured on May 31, 2019, and June 2, 2020, at the grape field of the Research Institute of Environment, Agriculture and Fisheries, Osaka Prefecture, as shown in Fig 1. In this field, grapes are grown in a horizontal trellis, which is popular for growing table grapes in Japan. The grape varieties were 'Fujiminori' grape (*Vitis vinifera* L. × *V. labruscana* Baily), 'Kyoho' grape (*Vitis vinifera* L. × *V. labruscana* Baily), 'Sun Verde' grape (*Vitis labruscana* Baily × *V. vinifera* L.) and 'Shine Muscat' grape (*Vitis labruscana* Baily × *V. vinifera* L.). The quantities of captured bunches were 127, 177, 107, and 153, respectively. The growth stage

when we captured images was the fruit set stage (E-L system 31-32) [27]. We used one of two lenses of a calibrated Richo THETA S to capture the videos. We moved the camera around each grape bunch to capture the whole bunch while trying not to move it. The original frame size of the videos captured was 960 × 960 pixels. The videos are shown in [28].

## Method overview

We describe how the 3D positions of grape berries can be estimated with our proposed methods in the following sections. First, in **Unsupervised monocular depth estimation**, we describe how unsupervised monocular depth estimation works and its extension to a Unified Omnidirectional Camera Model (UOCM). Thanks to the depth estimation using deep neural networks, 3D structures of grape bunches and camera poses can be estimated well only by preparing unlabeled videos of grape bunches. Next, in **Estimation of 3D Positions of Grape Berries**, we present how the estimation of the 3D positions of grape berries can be adjusted by bundle adjustment, which simultaneously optimizes the 3D positions of grape berries and camera poses of all video frames. Before applying bundle adjustment, grape berries must be matched over frames, and we explain how they can be tracked with our proposed methods described in **Grape Berry Matching**.

## Unsupervised monocular depth estimation

**Stereo vision basics.** We begin by presenting the basics of stereo vision based on the pinhole camera model, which is necessary to understand the unsupervised monocular depth estimation and our proposed method. As shown in Fig 3(a), stereo methods estimate the depth of objects using the disparity between the two images captured by two cameras. Disparity is the difference in the positions in which objects appear in a pair of stereo images. The distance between two cameras is called the baseline. Although image planes are placed inside cameras (behind camera centers, to be precise), they are often drawn in front of cameras for convenience. They are called virtual image planes. The virtual image planes are mathematically equivalent to the real image planes. The (virtual) image planes are placed at the distance of the focal length. We obtain the depth at a point on the object by using two similar triangles as

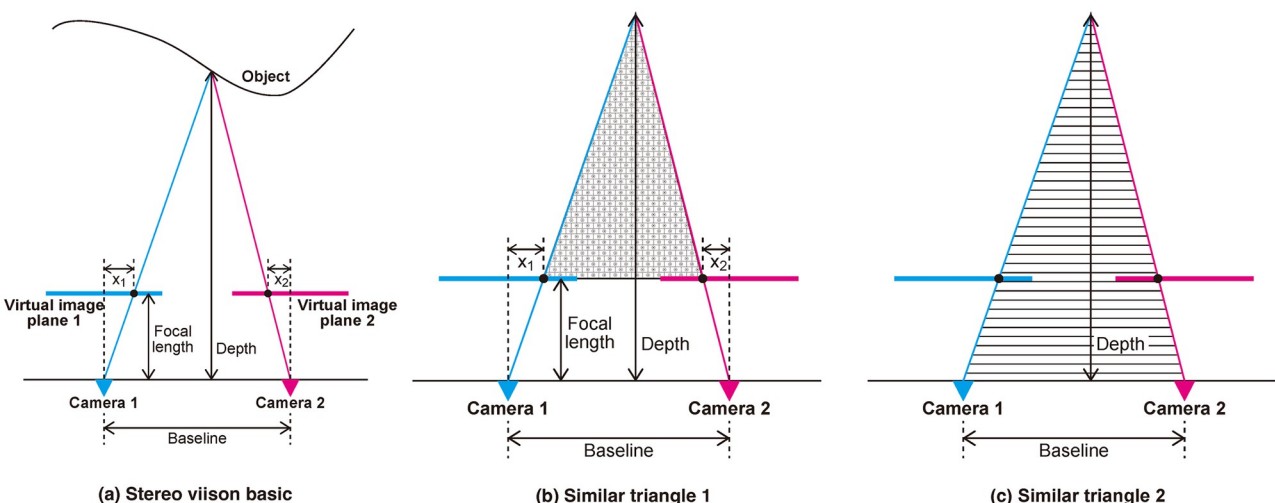

**Fig 3. Overview of the stereo vision.** Best viewed in color.

shown in Fig 3(b) and 3(c) as

$$\frac{(\text{Baseline}) - X_1 - X_2}{(\text{Depth}) - (\text{Focal length})} = \frac{(\text{Baseline})}{(\text{Depth})}. \tag{1}$$

Doing this at every pixel on the (virtual) image planes, we obtain the depth maps, which are images whose pixel values represent depth.

Fig 3 assume two-camera cases thus far. However, stereo vision can be realized with a single camera. For example, in the case that cameras 1 and 2 in the figures correspond to the single camera at times $t$ and $t + 1$, respectively. Monocular depth estimation is depth estimation using a single camera.

**Overview.**   As seen in Fig 3(a), a point on the objects is mapped in two image planes. In passive stereo methods, the corresponding points in two image planes should be identified to estimate the depth of the point. This is not easy to do on textureless surfaces. Therefore, neural network-based approaches that do not rely on corresponding points are worthwhile. In this way, models learn a mapping function from an image to its depth map. Therefore, input images and their depth maps as supervision are required to train models. However, true depth maps are not easy to obtain in real environments.

Therefore, a method that does not require true depth maps for training models, unsupervised monocular depth estimation, is introduced. Instead of true depth maps, the method uses a reasonable constraint that the pixel values of the points on the objects should be shared on both image planes. For example, a salient point on an object is projected onto both image planes, so they share the same pixel value. We can use this constraint to reconstruct one image (e.g., image 2) from the other image (e.g., image 1) without knowing the pixel values on the objects. Suppose that we know the relative camera pose between two cameras and the depth map of camera 1 correctly (see Fig 3(a) for reference). Then, we can predict which pixel in image 2 corresponds to a pixel in image 1. By assuming that the corresponding pixels share the same pixel values, we can reconstruct image 2 from image 1 without knowing 3D information of the objects. Intuitively, as long as the relative camera pose and the depth map of camera 1 are correct, image 2 and reconstructed image 2 must be the same. However, if the relative camera pose and the depth map are less correct, there should be an error between the pixel values of the two images (i.e., the original and reconstructed image 2). Therefore, we can update the model's network weights to minimize the error between the two images without the true depth maps. This process is called depth-image-based rendering (DIBR) [19]. By making it differentiable, we can use it for backpropagation [29].

As shown in Fig 4(a), the target frame $I_t$ is the video frame at time step $t$. Similarly, the source frame $I_{t'}$, where $t' \in \{t - 1, t + 1\}$, is an adjacent frame to the target frame. This setup defines the temporal context for the frames being analyzed and processed. The *depth network* plays a crucial role in this framework by taking only the target frame as input and estimating its depth map $D_t$. This estimation is crucial for understanding the scene structure and for subsequent processing steps. On the other hand, the *pose network*, which takes both the target and the source frames as input, estimates their relative camera pose, denoted as $T_{t \rightarrow t'}$, which encapsulates the movement and orientation changes between the two frames. It is important to note that while these networks are jointly trained to exploit the interdependencies between depth and pose for improved performance, they can be used independently during inference. This flexibility allows for various applications where either depth or pose estimation is required in isolation.

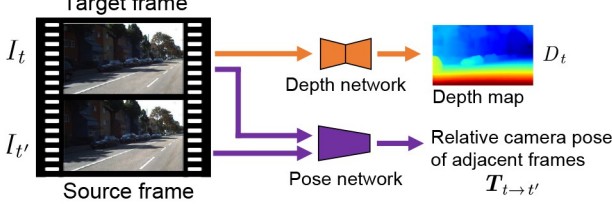

(a) Estimation of depth and pose.

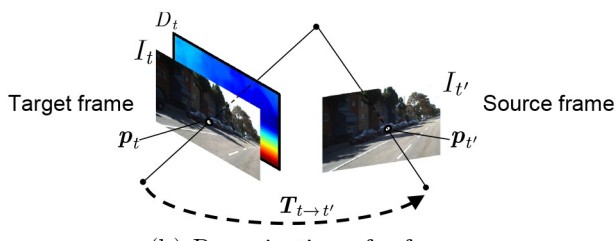

(b) Reprojection of a frame.

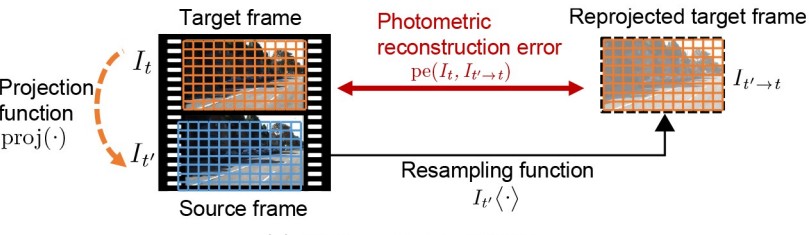

(c) Differentiable DIBR.

**Fig 4. Unsupervised monocular depth estimation using differentiable DIBR.**

**Formulation.** To train networks without ground truth depth data and camera poses, we consider projecting pixel values from one image onto the other. The underlying principle is that correct estimation of an image's depth map and the relative camera position between two images should result in minimal projection error. Therefore, during training, the network weights are updated to minimize this error.

As shown in Fig 4(c), let us denote the projection error (also known as photometric reconstruction error in DIBR) as $\mathrm{pe}(I_t, I_{t'\to t})$, where $I_{t'\to t}$ represents the reconstruction of $I_t$ from $I_{t'}$. Fig 4(b) illustrates that corresponding pixels are found in the source and target frames by using the depth map $D_t$ and the relative camera pose $\boldsymbol{T}_{t\to t'}$. As a consequence, a pixel $\mathbf{p}_t$ in the target frame $I_t$ corresponds to the pixel $\mathbf{p}_{t'}$ in the source frame $I_{t'}$.

The projection function $\mathrm{proj}(\cdot)$, shown in Fig 4(c), conducts this pixel-level mapping for all the pixels in the target frame $I_t$. That is, $\mathrm{proj}(\cdot)$ determines the mapping from the pixels in the grid of the target frame $I_t$ in orange to those of the source frame $I_{t'}$ in blue. Note that the input of $\mathrm{proj}(\cdot)$ is a set of 2D discrete coordinates, while the output is a set of 2D coordinates with continuous values. Thus, the mapped pixels are, for the most part, placed between regular discrete pixels in the grid of $I_{t'}$ and then interpolation is applied. More specifically, the output of the projection function is denoted by $\mathrm{proj}(D_t, \boldsymbol{T}_{t\to t'}, \boldsymbol{\theta}_{\mathrm{intrinsic}})$. That is, the projection function depends on a depth map $D_t$ and the relative camera pose $\boldsymbol{T}_{t\to t'}$. To be exact, it also depends on the vector $\boldsymbol{\theta}_{\mathrm{intrinsic}}$, composed of the intrinsic camera parameters. The intrinsic camera

parameters represent the optical center and focal length of the camera. The projection function maps all pixels in the target frame $I_t$ into the corresponding projected points in the source frame $I_{t'}$. Since the corresponding projected points are determined based on the camera model, the projection function depends on the camera model. In conventional studies, mainly a pinhole camera model is used, whereas we use an omnidirectional camera model.

The resampling function $I_{t'}\langle\cdot\rangle$ resamples the source frame $I_{t'}$ based on the output of the projection function. In Fig 4(c), it calculates the pixel values in the mapped grid using a method similar to bilinear interpolation. By means of this interpolation, the target frame is synthesized based on the mapped pixels and the source frame as reconstructed target frame $I_{t'\rightarrow t}$. In implementation, following [19], we adopt differentiable bilinear interpolation in spatial transformer networks [30], which normalize the sizes and positions of the input image, for the operation of the resampling function.

With the two functions above, the reconstructed target frame $I_{t'\rightarrow t}$ is calculated as

$$I_{t'\rightarrow t} = I_{t'}\langle\text{proj}(D_t, \boldsymbol{T}_{t\rightarrow t'}, \boldsymbol{\theta}_{\text{intrinsic}})\rangle. \tag{2}$$

The pixel-level differences between the original and reconstructed target frames, which is called the photometric reconstruction error, gives supervision for training the neural network models via error backpropagation. The photometric reprojection error of the target frame $I_t$ is formulated as the sum of photometric reconstruction errors over source frame $t' \in \{t-1, t+1\}$, which is denoted by

$$L_{\text{reprojection}} = \sum_{t'\in\{t-1,t+1\}} \text{pe}(I_t, I_{t'\rightarrow t}). \tag{3}$$

Following [20], pe($\cdot$) consists of the following two penalty terms. One is the L1 norm, which is a distance measure given in the form of $\|\cdot\|_1$. The other is the structural similarity index measure (SSIM) [31], which predicts the human-perceived quality of digital images and videos. Consequently, pe($I_a, I_b$) for two images $I_a$ and $I_b$ is given by

$$\text{pe}(I_a, I_b) = \frac{\alpha}{2}(1 - \text{SSIM}(I_a, I_b)) + (1-\alpha)\|I_a - I_b\|_1, \tag{4}$$

where $\alpha$ is a hyperparameter. We also add a smoothness loss term, following [20, 21].

To make the estimated depth maps reasonably smooth, we add the smoothness loss of the depth estimation, following [20, 21]. To do this, [21] introduced a normalized inverse depth map, denoted by $d_t^*$, to mitigate the divergence in depth estimation during training. The smoothness loss for the target frame $I_t$ is denoted by

$$L_{\text{smoothness}} = |\partial_x d_t^*|e^{-|\partial_x I_t|} + |\partial_y d_t^*|e^{-|\partial_y I_t|}, \tag{5}$$

where $\partial_x$ and $\partial_y$ denote differential along the $x$- and $y$-axes, respectively.

**Camera models.**   The projection function proj($\cdot$) is inherently dependent on camera models because it requires interactions between the corresponding 3D and 2D points through two operations, called projection and unprojection. The *projection* projects a 3D point onto the image planes, and the *unprojection* projects a 2D point in the image planes back into the 3D space. Though adopting a pinhole camera model is the most popular, we propose extending it to UOCM for our proposed pruning assisting system.

In the case of the pinhole camera model, as shown in Fig 5(a), a 3D point $\boldsymbol{x}_t = (x_t, y_t, z_t)^\top$ is projected onto the normalized image plane. The homogeneous coordinate $\tilde{\boldsymbol{p}}_t$ of the projected

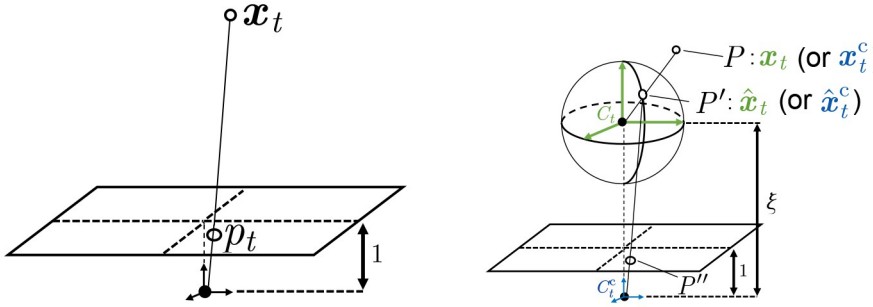

(a) Pinhole camera model.　　(b) Unified omnidirectional camera model.

**Fig 5. Comparison of camera models.** Coordinate systems of $C_t$ and $C_t^c$ are in green or blue, respectively.

2D image point $\boldsymbol{p}_t$ is calculated as

$$\tilde{\boldsymbol{p}}_t = \begin{pmatrix} \boldsymbol{p}_t \\ 1 \end{pmatrix} = \frac{1}{z_t} \boldsymbol{K} \boldsymbol{x}_t, \tag{6}$$

where $\boldsymbol{K}$ is a matrix consisting of the intrinsic camera parameters that make up the vector $\boldsymbol{\theta}_{\text{intrinsic}}$. Unprojection of a pixel $\boldsymbol{p}_t$ to its corresponding 3D point, $\boldsymbol{x}_t$, can be easily calculated from Eq (6).

The UOCM models cameras with a wide field of view, such as omnidirectional cameras. It was originally proposed in [32], and has been used in some 3D reconstruction [13] and depth estimation [33] studies utilizing omnidirectional cameras. Fig 5(b) illustrates an overview of the camera model, where the target frame $I_t$ is obtained using two successive projections: (1) a 3D point (say, $P$) is projected onto the surface of the unit sphere centered at $C_t$ (the projected point is denoted by $P'$), and then (2) projected onto the normalized plane of the camera center $C_t^c$ with a pinhole camera model (the projected point is denoted by $P''$). The coordinate system of $C_t$ is given by shifting that of $C_t^c$ by $\xi$ along its $z$ axis, where $\xi$ is a hyperparameter.

We start by formulating the projection of a 3D point onto the image plane and its inverse (i.e., unprojection [13]) in UOCM. We first consider the coordinate of a point in the coordinate system of $C_t$ and then convert it into the coordinate system of $C_t^c$ by adding $\boldsymbol{t}_\xi$ to the coordinate. Assume that the coordinate of the 3D point $P$ is given in the coordinate system of $C_t$ by $\boldsymbol{x}_t = (x_t, y_t, z_t)^\top$. Letting $\boldsymbol{t}_\xi = (0, 0, \xi)^\top$, the same point $P$ is denoted in the coordinate system of $C_t^c$ by

$$\boldsymbol{x}_t^c = \boldsymbol{x}_t + \boldsymbol{t}_\xi = (x_t, y_t, z_t + \xi)^\top. \tag{7}$$

Next, the coordinate of the projected point $P'$, denoted by $\hat{\boldsymbol{x}}_t$, is given in the coordinate system of $C_t$ by

$$\hat{\boldsymbol{x}}_t = \frac{\boldsymbol{x}_t}{\|\boldsymbol{x}_t\|} = \begin{pmatrix} x_t/\|\boldsymbol{x}_t\| \\ y_t/\|\boldsymbol{x}_t\| \\ z_t/\|\boldsymbol{x}_t\| \end{pmatrix}. \tag{8}$$

The same point is given in the coordinate system of $C_t^c$ by

$$\hat{\boldsymbol{x}}_t^c = \frac{\boldsymbol{x}_t}{||\boldsymbol{x}_t||} + \boldsymbol{t}_\xi = \begin{pmatrix} x_t/||\boldsymbol{x}_t|| \\ y_t/||\boldsymbol{x}_t|| \\ z_t/||\boldsymbol{x}_t|| + \xi \end{pmatrix}. \tag{9}$$

Then, $\hat{\boldsymbol{x}}_t^c$ is projected in the same manner as a pinhole camera onto the normalized image plane of $C_t^c$. To do that, we use the equations of pinhole projection. Substituting $\hat{\boldsymbol{x}}_t^c$ of Eq (9) into $\boldsymbol{x}_t$ of Eq (6), and also substituting $z_t/||\boldsymbol{x}_t|| + \xi$, which is the $z$ coordinate of $\hat{\boldsymbol{x}}_t^c$, into $z_t$ of Eq (6), the projection of UOCM is computed as

$$\tilde{\boldsymbol{p}}_t = \frac{1}{z_t/||\boldsymbol{x}_t|| + \xi} \boldsymbol{K}\hat{\boldsymbol{x}}_t^c = \frac{1}{z_t + \xi||\boldsymbol{x}_t||} \boldsymbol{K}(\boldsymbol{x}_t + \boldsymbol{t}_\xi||\boldsymbol{x}_t||). \tag{10}$$

In addition, the image point $\boldsymbol{p}_t$ is unprojected to the coordinate system of $C_t$ in a closed form, denoted by

$$\frac{\boldsymbol{x}_t}{||\boldsymbol{x}_t||} = \frac{\xi + \sqrt{1 + (1 - \xi^2)(||\boldsymbol{K}^{-1}\tilde{\boldsymbol{p}}_t||^2 - 1)}}{||\boldsymbol{K}^{-1}\tilde{\boldsymbol{p}}_t||^2} \boldsymbol{K}^{-1}\tilde{\boldsymbol{p}}_t - \boldsymbol{t}_\xi. \tag{11}$$

Note that the coordinate of the 3D point $P$ is denoted by $\frac{\boldsymbol{x}_t}{||\boldsymbol{x}_t||}$ because the unprojection has ambiguity in estimating the depth of the point from its nature.

**Network architectures.** We employed the same network architectures for both the depth and pose networks as those described in [20].

For single-view depth prediction, we utilize the DispNet architecture introduced in [34]. This architecture is primarily based on an encoder-decoder framework that incorporates skip connections and multi-scale side predictions. All convolutional layers are followed by ReLU activation, except for the prediction layers. In these layers, we apply $1/(\alpha * \text{sigmoid}(x) + \beta)$, where $\alpha = 10$ and $\beta = 0.01$, ensuring the predicted depth remains positive and within a realistic range.

For the pose network, the input consists of the target view concatenated with all source views (along the color channels), and the output comprises the relative poses between the target view and each of the source views. The network architecture includes seven stride-2 convolutional layers, followed by a $1 \times 1$ convolution layer with $6 * (N - 1)$ output channels, representing the 3 Euler angles and 3D translations for each source view. Finally, global average pooling is applied to aggregate predictions at all spatial locations. All conv layers are followed by ReLU except for the last layer where no nonlinear activation is applied.

We implemented the proposed unsupervised monocular depth estimation method based on the implementation of [20]. The hyperparameters were set to the default values provided in the implementation of [20]. Specifically, the parameters $d_{\min}$ and $d_{\max}$ were set to 1 and 100, respectively. The networks were trained for ten epochs using an NVIDIA TITAN Xp GPU.

## Estimation of 3D positions of grape berries

In this section, we explain how to estimate the 3D positions of the berries in a grape bunch from a video using the proposed monocular depth estimation method. To apply the method to the omnidirectional camera, adopted in our pruning support system, we explain inter-frame matching of grape berries and bundle adjustment.

**Overview.** The ultimate goal of our research is to estimate the 3D arrangement of the grape berries of a whole bunch, as illustrated in Fig 2(b). To this end, in this paper, we propose

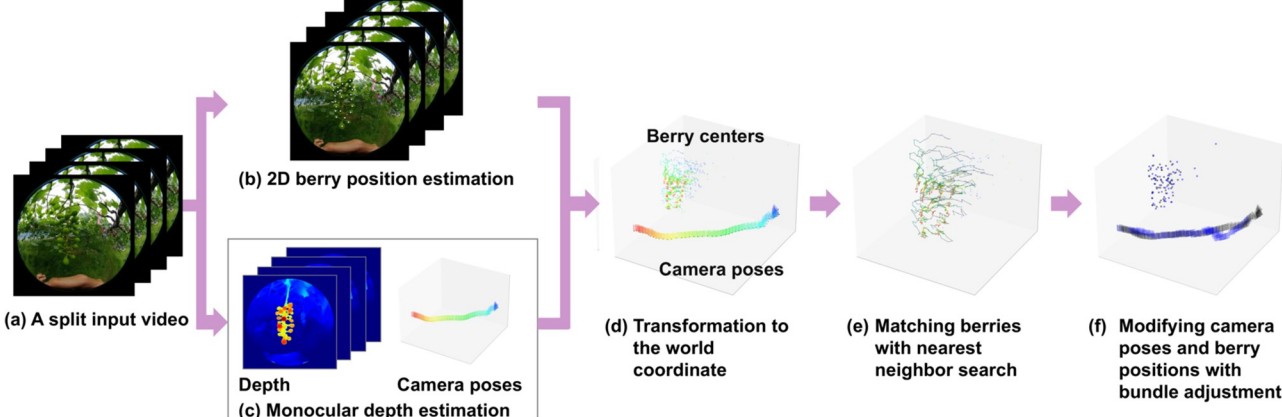

**Fig 6. Flow of estimation berry positions of grape bunches.**

a method for estimating the berry positions of a grape bunch. An overview of this estimation method is illustrated in Fig 6. As shown in Fig 6(a), we divide the full video frames into smaller parts and input only a partial video each time. This division strategy is adopted because errors in camera poses estimated by the pose network accumulate over frames and lead to failure in estimating the 3D positions of grape berries. Given a split input video, the 2D positions of grape berry centers in the frames are estimated, as displayed in Fig 6(b). We estimate the 2D positions of grape berry centers using an instance segmentation method [35], although the module is not yet unified in the proposed method. In parallel, as shown in Fig 6(c), the depths and relative camera poses of the frames are sequentially estimated using the proposed unsupervised monocular depth estimation methods, presented in **Unsupervised monocular depth estimation**. Based on the estimation results, the 3D positions of berry centers are estimated in the coordinate system of each camera $C_t$, the camera center at a time step $t$. The estimated 3D positions of berry centers are unprojected to the world coordinate, as visualized in Fig 6(d). As shown in Fig 6(e), those unprojected berries are matched across frames using nearest neighbor search, which is a technique that finds the closest point from the query point. Finally, as shown in Fig 6(f), the arrangement of the grape berries and the relative camera poses are simulateneously modified with bundle adjustment in a partial video. The result of the bundle adjustment is then merged over the full video frames to obtain the 3D information of a whole grape bunch, as illustrated in Fig 2(b).

**Grape berry matching.** As a preparation for the bundle adjustment presented in **Bundle Adjustment**, we need to find the correspondences of the berries across the frames. More precisely, we match grape berry centers of different frames using the 3D positions of the berry centers, their depths, and the camera poses of consecutive frames.

Assume that there are $K_t$ feature points on the frame of camera $C_t$. The 3D position of the $k_t$-th feature point in the coordinate system of camera $C_t$, $\boldsymbol{x}_t^{(k_t)}$ $(k_t = 1, \ldots, K_t)$, is estimated by unprojecting the $k_t$-th feature point from the image plane to the camera coordinate of $C_t$ with the estimated depth. Let $\boldsymbol{w}_t^{(k_t)}$ be the coordinate of $\boldsymbol{x}_t^{(k_t)}$ in the world coordinate system, then $\boldsymbol{w}_t^{(k_t)}$ is calculated as

$$\begin{pmatrix} \boldsymbol{w}_t^{(k_t)} \\ 1 \end{pmatrix} = \boldsymbol{T}_t^{-1} \cdot \tilde{\boldsymbol{x}}_t^{(k_t)} = \boldsymbol{T}_t^{-1} \cdot \begin{pmatrix} \boldsymbol{x}_t^{(k_t)} \\ 1 \end{pmatrix}, \tag{12}$$

where $\boldsymbol{T}_t = \Pi_{l=1}^{t} \boldsymbol{T}_{l \to l+1}$, which is the relative camera pose from the first frame to the $t$-th frame. Fig 6(d) shows examples of the estimated 3D positions of feature points. The colors of the points indicate their corresponding cameras.

As mentioned above, we use only the 3D coordinates of feature points to match and group berries across frames. For robust matching of the coordinates, we use nearest neighbor search in both forward and backward directions along time. The forward search finds the corresponding points of the current frame in the next frame, while the backward search does the opposite. Then, only the matched points in both searches are grouped. When the number of sequentially corresponding points in a set is bigger than a threshold (we use five in this study), the set is used for bundle adjustment.

**Bundle adjustment.** Bundle adjustment optimizes camera poses and the 3D positions of feature points such that they minimize reprojection errors. In general, reprojection errors are calculated with the residuals between 2D feature points and the corresponding reprojected 3D feature points in each frame. For better convergence of the optimization, we calculate residuals on each unit sphere whose center is $C_t$, following [33], instead of calculating residuals in each frame.

We then formulate bundle adjustment with UOCM. Let $N_I$ be the number of cameras optimized in bundle adjustment, and $C_t$ be the center of the unit sphere corresponding to camera $t$, where $t = 1, \ldots, N_I$. The right image in Fig 6(c) shows an example of a set of consecutive cameras. Let us regard the camera coordinate system of $C_1$ as the world coordinate system. The extrinsic parameter $\boldsymbol{T}_t$ contains its rotation matrix $\boldsymbol{R}_t$ and translation vector $\boldsymbol{t}_t$. Note that $\boldsymbol{T}_1$ is the identity matrix. Let $\boldsymbol{w}_j$ and $\boldsymbol{x}_{t,j}$ be the 3D coordinates of the $j$-th feature point in the world coordinate system and the coordinate system of $C_t$, respectively. Then, bundle adjustment in UOCM is given as a minimization problem by

$$\underset{\{\boldsymbol{w}_j\}, \{\boldsymbol{R}_t\}, \{\boldsymbol{t}_t\}}{\mathrm{argmin}} \sum_{t,j} ||\langle \boldsymbol{x}_{t,j} \rangle - \langle \boldsymbol{R}_t \cdot \boldsymbol{w}_j + \boldsymbol{t}_t \rangle||_{\mathrm{H}}, \tag{13}$$

where $||\cdot||_{\mathrm{H}}$ is Huber loss for robust optimization against outliers. The initial values of $\boldsymbol{w}_j$ come from the mean of the $j$-th set of sequentially corresponding feature points obtained in **Grape Berry Matching**. The initial values of $\boldsymbol{T}_t$ are the same as the ones in Eq (12).

Optimization of Eq (13) suffers from ambiguity of scale. The scales of the camera poses and feature points change substantially depending on choices of $C_1$. To mitigate the scale ambiguity problem, we alternately fix the feature points and the camera poses to calculate Eq (13).

## Results

We conducted an experiment of depth estimation of grape bunches, and with its results we estimated positions of grape berries by bundle adjustment.

### Experimental setup

The frame size of the videos used for the experiment was 480×480 pixels, and the frame rate was 30 frames per second (fps). The frames were resized from the original videos shown in **Materials** by nearest neighbor interpolation for lower memory consumpton and higher computational speed. We captured one video per bunch; the average length of the videos was less than 20 seconds. The total number of videos in the dataset was 564. For training the networks, we used randomly selected frames from the dataset, except for test videos. The numbers of frames used for training and validation were 128,068 and 31,554, respectively.

## 3D shape estimation of grape berries

For depth estimation evaluation, we used one bunch each of Fujiminori and Shine Muscat, which were excluded from the training data. As shown in Fig 7, we partially estimated the berry positions of two grape bunches in the test dataset using videos of Fujiminori and Shine Muscat, composed of about 200 and 300 frames, respectively. Fig 7(a) are from a video of Fujiminori, and Fig 7(b) are from a video of Shine Muscat. The rows of Fig 7 show the first, 50th, 100th, and 150th frames of each video, from top to bottom. The heatmaps presented in the second column of Fig 7(a) and 7(b) show inverse depths of input frames in the first column of Fig 7(a) and 7(b); the closer objects are to the camera, the redder the heat maps are. Inverse depths outside of the imaging range were set to 0. The third and fourth columns of Fig 7(a) and 7(b) show point clouds seen from two angles, generated with estimated depths and input RGB pixels.

## Results of bundle adjustment

We conducted two types of bundle adjustment experiments: optimizing the positions of berries $w_j$ and the camera poses $T_i$ of Eq (13) simultaneously, and optimizing them alternately. We used the same video of the Fujiminori bunch used to evaluate depth estimation. We manually plotted the 2D positions of berry centers in the frames. The plots of camera poses and berry positions after bundle adjustment, applied to the first to 50th frames of the video, are shown in Fig 8. Fig 8(a) shows the result of simultaneous optimization. In this case, bundle adjustment caused considerable scale changes. In contrast, as Fig 8(b) shows, when the camera poses and berry positions are optimized alternately, the scales are mostly preserved. Therefore, our results show that the alternate optimization shown in Fig 8(b) is better for preserving the scale.

## Discussion

In **3D Shape Estimation of Grape Berries**, experimental results of Fig 7 show that 3D structures of grape bunches can be partially estimated well even though textureless and homogeneous crowded berries are placed around them. Our approach of using unsupervised monocular depth estimation is also effective for objects in environments outside where lighting is instable. Another advantage of our method is that it only needs RGB frames taken with a normal video camera, as long as its camera model is known. In other words depth information of objects of interest can be estiamted only with an RGB camera, without a depth sensor.

In **Result of Bundle Adjustment**, we have also shown that those estimation accuracies of grape positions can be improved by matching berries over frames and bundle adjustment in Fig 8. The results also imply that 3D positions of all berries in a whole bunch can be estimated by merging the final results presented in our method.

The experimental environment for this research is no different from a typical Japanese field of fresh grapes. The bunches hanging from the trellis-trained grape vines were captured and used in the experiment. Therefore, the experiment was carried out under realistic conditions, and the fact that 3D reconstruction was successful in this environment shows that the proposed method is sufficiently robust for practical use.

The limitations of our method are that some tasks cannot be automated and must be done manually. One process we have yet to automate is estimating the centers of grape berries in each frame. As [36] shows and as we empirically know, each grape berry can be segmented with recent instance segmentation methods such as Mask R-CNN [35]. After segmenting each berry, the center of each berry can be easily calculated. It is noteworthy that instance segmentation models pre-trained with a large benchmark dataset needs relatively few additional labeled

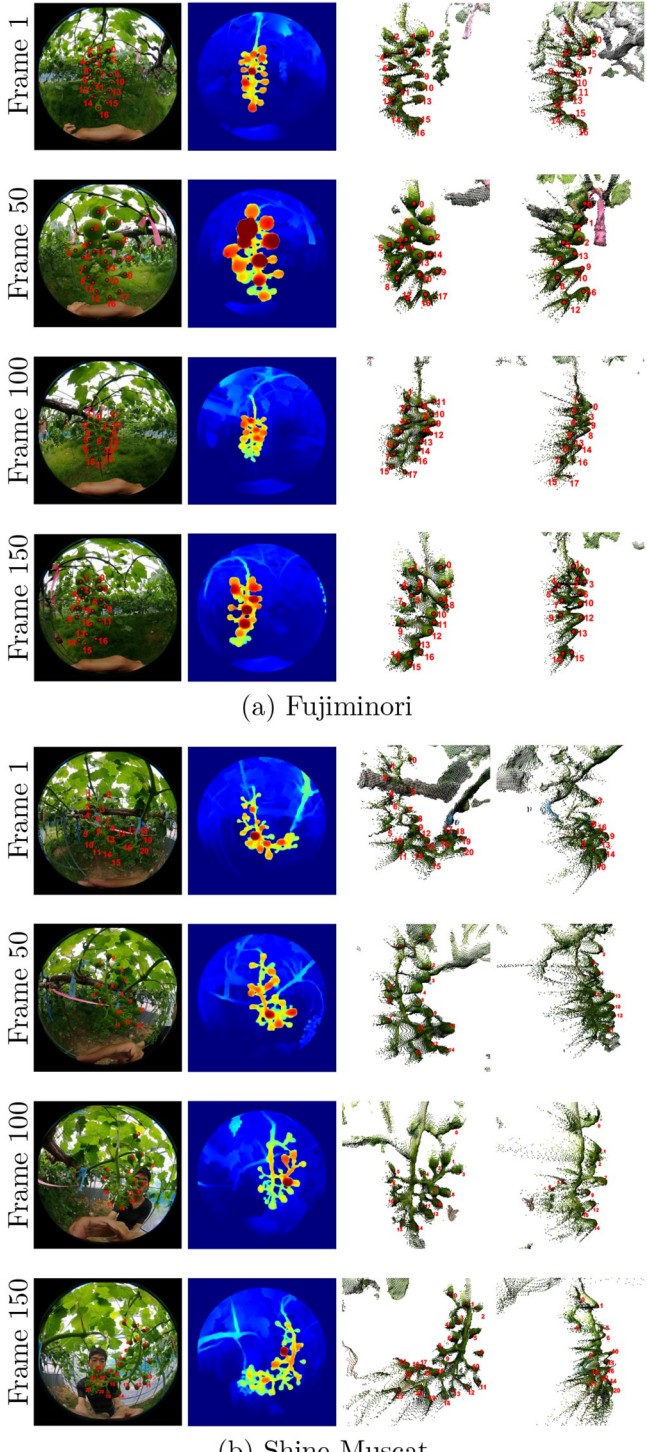

**Fig 7. Estimated 3D shapes of grape berries.** (a) and (b) show different bunches. First, second, third and forth columns are input, inverse depth estimation results (in Jet colormap, namely the closer the redder), generated point cloud (diagonal view), and generated point cloud (side view), respectively. Regions closer than a certain degree are converted to point clouds. Red points and numbers are manual annotations which show berry correspondence.

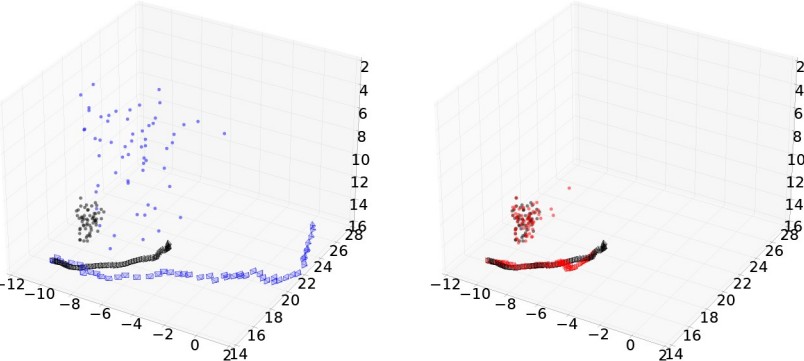

(a) Simultaneous optimization.     (b) Alternate optimization.

**Fig 8. Camera poses and berry positions after bundle adjustment.** Points and pyramids in black indicate initial positions and poses of berries and cameras. Those in blue or red indicate results after bundle adjustment.

images for segmentations of each grape berry. Therefore, training the neural network models of our system requires a limited amount of human supervision only on how to segment each berry.

Another limitation of the proposed method is the calculation time. Since the combinations of methods introduced so far, such as deep neural networks and bundle adjustment, can be computationally expensive. Adopting a client-server system between a user and a remote server is an option. In this case, all a user needs is an omnidirectional camera, an Internet connection, and a device to display supervision on pruning. The use of wearable devices can be considered for displaying the results of the automated selection of berries to cut.

The challenges in applying this method in practice include variations in the growth stages of grape bunches, differences in image appearance due to field conditions, and changes in lighting caused by weather. In agriculture, these variations often hinder the effectiveness of image processing. Our proposed method addresses these issues by using additional training to adapt to the differences. Specifically, the monocular depth estimation used in the proposed method can be trained using videos recorded without the need for ground truth data, making additional training easy. As a result, the proposed method effectively handles these variations with ease.

Estimating 3D arrangement of grape berries have advantages when it comes to supporting table grape pruning. Grape pruning has several rules such as a total number of berries or distances between adjacent berries. Conventionally, selection of berries has been conducted based on farmers' experiences. However, with the information on the 3D positions of grape berries, berries for removal can be selected by optimizing desirable arrangement of berries without direct supervision by skilled farmers. The development of the technology has the potential to enable individuals without prior experience in grape cultivation to actively participate in the process. This could facilitate the recruitment of a broader labor force more efficiently than before, thereby supporting the expansion of cultivation areas. Furthermore, the technology is anticipated to advance the automation of pruning tasks, which is likely to contribute to a significant reduction in grape production costs.

## Conclusion

This project aimed to estimate the 3D arrangement of whole bunches of grape berries in the wild, with the ultimate goal of supporting the mastering of the pruning task needed for

Japanese table grape cultivation. Due to some restrictions we faced, and for convenience, we chose to use videos captured with an omnidirectional camera. To realize the estimation of the 3D positions of grape berries, we extended the differentiable DIBR to omnidirectional cameras. To this end, we estimated the 3D structure of grape bunches in the wild using only an omnidirectional camera, without any depth sensor or labeled data. To extend the result of the 3D estimation from per-frame to sequential frames, we introduced bundle adjustment for textureless objects to a video captured with an omnidirectional camera. To mitigate the problem of scale instability, we alternately optimized camera poses and berry positions.

Our experiments demonstrated that the proposed method can successfully estimate the depth of grape berries and generate a point cloud of grapes per frame. In addition, we succeeded in matching berries using partial videos of grapes by the proposed methods, thereby estimating the arrangement of grape berries without changing their scales. Thus, it is expected that they can be relatively easily merged as whole bunches in our future work.

## Author Contributions

**Conceptualization:** Yuzuko Utsumi, Yuka Miwa, Masakazu Iwamura.

**Data curation:** Yuka Miwa.

**Formal analysis:** Yasuto Tamura.

**Funding acquisition:** Yuzuko Utsumi.

**Investigation:** Yasuto Tamura.

**Methodology:** Yasuto Tamura.

**Project administration:** Yuzuko Utsumi, Masakazu Iwamura.

**Resources:** Koichi Kise.

**Software:** Yasuto Tamura.

**Supervision:** Yuzuko Utsumi, Masakazu Iwamura, Koichi Kise.

**Validation:** Yasuto Tamura.

**Visualization:** Yasuto Tamura.

**Writing – original draft:** Yasuto Tamura.

**Writing – review & editing:** Yasuto Tamura, Yuzuko Utsumi, Yuka Miwa, Masakazu Iwamura, Koichi Kise.

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
