## [Decision Letter · Decision Letter 0]

10 Sep 2024

PONE-D-24-25787Unsupervised Monocular Depth Estimation with Omnidirectional Camera and Its Application to Obtaining 3D Structure of Textureless, Symmetric and Crowded Grape Berries in the WildPLOS ONE

Dear Dr. Utsumi,

Thank you for submitting your manuscript to PLOS ONE. After careful consideration, we feel that it has merit but does not fully meet PLOS ONE’s publication criteria as it currently stands. Therefore, we invite you to submit a revised version of the manuscript that addresses the points raised during the review process.

**The reviewers raised comments that need to be addressed by the authors.**

We look forward to receiving your revised manuscript.

Kind regards,

Alberto Marchisio

Academic Editor

PLOS ONE

**Journal Requirements:**

Reviewers' comments:

Reviewer's Responses to Questions

**Comments to the Author**

1. Is the manuscript technically sound, and do the data support the conclusions?

Reviewer #1: Yes

Reviewer #2: Yes

Reviewer #3: Yes

Reviewer #4: Yes

Reviewer #5: Yes

2. Has the statistical analysis been performed appropriately and rigorously? 

Reviewer #1: No

Reviewer #2: N/A

Reviewer #3: Yes

Reviewer #4: Yes

Reviewer #5: Yes

3. Have the authors made all data underlying the findings in their manuscript fully available?

Reviewer #1: No

Reviewer #2: Yes

Reviewer #3: Yes

Reviewer #4: Yes

Reviewer #5: Yes

4. Is the manuscript presented in an intelligible fashion and written in standard English?

Reviewer #1: Yes

Reviewer #2: No

Reviewer #3: Yes

Reviewer #4: Yes

Reviewer #5: Yes

5. Review Comments to the Author

**Reviewer #1: **This paper proposes a unsupervised monocular depth estimation with omnidirectional camera and its application to obtaining 3d structure of textureless, symmetric and crowded grape berries in the wild

1- The structure of the paper should seriously be improved:

a. The introduction Section is not written to present the importance of the solved problem. I was confused. What is the main problem /s that you have solved? Please reduce the title of paper length.

b. The related works should be discussed and the prons and coins should be expressed in one paragraph in the introduction Section to make the proposed method/s’s efficient.In related work section, please add new references that are published in state-of-the-art. In addition. The logic of the mentioned references needs to be strengthened to highlight the innovations of this paper. The difficulty encountered in this research should be discussed in more detail. Please compare and add this references:

• Car depth estimation within a monocular image using a light CNN

• A New Approach To Estimate Depth Of Cars Using A Monocular Image

• An Efficient Approach to Monocular Depth Estimation for Autonomous Vehicle Perception Systems

2- The paper should be provided some realistic scenario with noises... to access the performanceof proposed approach.

Overall, the paper is written in good English and style. However, there are some typos as

3-well as grammatical errors. Could the authors please revise and proof-read the paper carefully

before resubmission for publication?

**Reviewer #2:** Please find below specific suggestions for improvements and corrections.

1. It would be better to add the recorded field video either in MS or Supplementary for the readers.

2. Some of the abbreviations have been used in the MS, which need to be written as a full name at the beginning and later can be used in the abbreviated forms.

3. Add some more detail to the discussion sections, please.

4. Abstract. Write “arrangements” instead of “arrangement”.

5. Abstract. rewrite “labor-intensive” instead of “labor intensive”.

6. Lines 1-3, kindly provide reference/source.

7. Lines 4-6, kindly provide reference/source.

8. Figures 2 and 3 are repetitions of the same phenomena (stereo vision), which need to be presented in a single figure having subsections a, b, and c.

9. Line 243, 360 and 374 ”fig. 5(B).”, fig. 7 (A) and fig. 8 (A) need to be “Fig. 5(B)”, “Fig. 7 (A)” and Fig. 8 (A) respectively.

**Reviewer #3:** Unsupervised Monocular Depth Estimation with Omnidirectional Camera and Its Application to Obtaining 3D Structure of Texture less, Symmetric and Crowded Grape Berries in the Wild

1. No keywords.

2. I suggest change the title of figure 1 to (Proposed grape pruning support system).

3. The paper lacks of literature reviews.

**Reviewer #4:** General comment:

The manuscript presents an innovative approach to addressing a practical problem in agricultural automation, specifically grape pruning. The utilization of unsupervised monocular depth estimation in conjunction with an omnidirectional camera for 3D reconstruction of grape bunches is both novel and relevant to the field. The methodology is sound, and the experimental results convincingly demonstrate the effectiveness of the proposed approach in estimating the 3D positions of grape berries in real-world conditions.

Areas for Improvement:

1. The manuscript lacks a detailed description of the neural network architecture used for depth and pose estimation. For instance, it does not specify the number of convolutional layers, kernel sizes, activation functions, or the overall architecture of the encoder-decoder framework. Providing this information is crucial for enabling replication and deeper understanding of the proposed method by the research community.

2. While the proposed method is effective under certain conditions, the manuscript could benefit from a more thorough discussion of how varying environmental factors, such as changes in lighting conditions and the different growth stages of grape berries, might affect the performance of the system. These factors are critical in agricultural applications, where conditions can be highly variable.

3. It would be valuable for the authors to clarify whether the primary objective of this research is to assist human workers by providing a consistent standard for grape pruning, or if it is intended as a preparatory step toward full automation of the pruning process. Each objective might require different approaches in terms of system design and evaluation criteria.

4. Conduct more real environment tests and present these results in the paper. This should include realistic scenarios for testing system performance under various conditions, highlighting its strengths and areas for improvement.

**Reviewer #5: **The manuscript presents technical and scientific merits about: Unsupervised Monocular Depth Estimation with Omnidirectional Camera and Its Application to Obtaining 3D Structure of Textureless, Symmetric and Crowded Grape Berries in the Wild. The Manuscript falls within the scope of the Plos One. However, from my expertis it is necessary to detail certain points so that the audience has a better understanding.

6. PLOS authors have the option to publish the peer review history of their article (what does this mean?). If published, this will include your full peer review and any attached files.

Reviewer #1: No

Reviewer #2: **Yes: **Jingyang Li

Reviewer #3: No

Reviewer #4: No

Reviewer #5: No

---

## [Author Response · Author response to Decision Letter 0]

13 Dec 2024

Thank you for taking the time to review this. We have attached a reply letter, so please find it.

---

## [Decision Letter · Decision Letter 1]

27 Dec 2024

Unsupervised Monocular Depth Estimation with Omnidirectional Camera for 3D Reconstruction of Grape Berries in the Wild

PONE-D-24-25787R1

Dear Dr. Utsumi,

We’re pleased to inform you that your manuscript has been judged scientifically suitable for publication and will be formally accepted for publication once it meets all outstanding technical requirements.

Kind regards,

Alberto Marchisio

Academic Editor

PLOS ONE

Additional Editor Comments (optional):

Reviewers' comments:

Reviewer's Responses to Questions

**Comments to the Author**

1. If the authors have adequately addressed your comments raised in a previous round of review and you feel that this manuscript is now acceptable for publication, you may indicate that here to bypass the “Comments to the Author” section, enter your conflict of interest statement in the “Confidential to Editor” section, and submit your "Accept" recommendation.

Reviewer #1: All comments have been addressed

Reviewer #2: All comments have been addressed

Reviewer #4: All comments have been addressed

Reviewer #5: All comments have been addressed

2. Is the manuscript technically sound, and do the data support the conclusions?

Reviewer #1: Yes

Reviewer #2: Partly

Reviewer #4: Yes

Reviewer #5: Yes

3. Has the statistical analysis been performed appropriately and rigorously? 

Reviewer #1: Yes

Reviewer #2: Yes

Reviewer #4: Yes

Reviewer #5: Yes

4. Have the authors made all data underlying the findings in their manuscript fully available?

Reviewer #1: Yes

Reviewer #2: Yes

Reviewer #4: Yes

Reviewer #5: Yes

5. Is the manuscript presented in an intelligible fashion and written in standard English?

Reviewer #1: Yes

Reviewer #2: Yes

Reviewer #4: Yes

Reviewer #5: Yes

6. Review Comments to the Author

Reviewer #1: (No Response)

Reviewer #2: 1. "estiamted" should be "estimated".

2. The sentence "Japanese table grapes are quite expensive because their production is highly labor - intensive." could be rephrased as "Japanese table grapes are considerably expensive due to their highly labor - intensive production process."

3. "Pruning berries is carried out at an early stage of grape growth, after young berries have borne, to produce an optimal spacing between berries to accommodate further growth." This sentence is a bit wordy and could be simplified to "Pruning berries is performed in the early growth stage after young berries have formed to create optimal spacing for further growth."

Reviewer #4: (No Response)

Reviewer #5: My observations have been addressed by the authors of the manuscript and have been incorporated into it.

7. PLOS authors have the option to publish the peer review history of their article (what does this mean?). If published, this will include your full peer review and any attached files.

Reviewer #1: **Yes: **zahra shirmohammadi

Reviewer #2: No

Reviewer #4: No

Reviewer #5: No

---

## [Editor Report · Acceptance letter]

24 Jan 2025

PONE-D-24-25787R1 

PLOS ONE

Dear Dr. Utsumi, 

I'm pleased to inform you that your manuscript has been deemed suitable for publication in PLOS ONE. Congratulations! Your manuscript is now being handed over to our production team.

Kind regards, 

on behalf of

Dr. Alberto Marchisio 

Academic Editor

PLOS ONE